# An Eight-Direction Scanning Detection Algorithm for the Mapping Robot Pathfinding in Unknown Indoor Environment

**DOI:** 10.3390/s18124254

**Published:** 2018-12-04

**Authors:** Le Jiang, Pengcheng Zhao, Wei Dong, Jiayuan Li, Mingyao Ai, Xuan Wu, Qingwu Hu

**Affiliations:** 1School of Remote Sensing & Information Engineering, Wuhan University, Wuhan 430079, China; jiangl@whu.edu.cn (L.J.); ljy_whu_2012@whu.edu.cn (J.L.); aimingyao@whu.edu.cn (M.A.); 2School of Physics and Technology, Wuhan University, Wuhan 430079, China; 3China Railway Siyuan Survey and Design Co., Ltd., Wuhan 430063, China; tsygkydw@163.com (W.D.); crfsdi_wx@163.com (X.W.)

**Keywords:** eight-direction scanning detection, unknown indoor environment, mapping robot, pathfinding, autonomous obstacle avoidance

## Abstract

Aiming at the problem of how to enable the mobile robot to navigate and traverse efficiently and safely in the unknown indoor environment and map the environment, an eight-direction scanning detection (eDSD) algorithm is proposed as a new pathfinding algorithm. Firstly, we use a laser-based SLAM (Simultaneous Localization and Mapping) algorithm to perform simultaneous localization and mapping to acquire the environment information around the robot. Then, according to the proposed algorithm, the 8 certain areas around the 8 directions which are developed from the robot’s center point are analyzed in order to calculate the probabilistic path vector of each area. Considering the requirements of efficient traverse and obstacle avoidance in practical applications, the proposal can find the optimal local path in a short time. In addition to local pathfinding, the global pathfinding is also introduced for unknown environments of large-scale and complex structures to reduce the repeated traverse. The field experiments in three typical indoor environments demonstrate that deviation of the planned path from the ideal path can be kept to a low level in terms of the path length and total time consumption. It is confirmed that the proposed algorithm is highly adaptable and practical in various indoor environments.

## 1. Introduction

Mobile robot is a comprehensive system integrating environment perception, dynamic decision and planning, behavior control, and execution, etc. [1]. Since the late 1960s, much research has been conducted on mobile robots’ environmental information sensors, information processing algorithms, remote control technologies, and navigation in the environment [2]. Nowadays, the application of mobile robots have been expanded from artificial intelligence platforms in the university laboratories or institutes to people’s everyday lives. In fact, mobile robots are widely applied in military, aviation, resource exploration, transportation, agriculture, and education, and have broad prospects [3,4,5,6], even in one of the most active areas in science and technology.

In the 3D data acquisition in indoor environments, people often use trolley or handheld devices [7]. However, since the equipment (panoramic camera) should be operated by researchers, the photos obtained will inevitably include them. Thus, the environmental information which they want to obtain will be blocked and they have to take more photos to eliminate the effect. This will cause great trouble to the researchers. In order to overcome the inconvenience, we intend to make autonomous robots serve as intelligent panoramic acquisition platforms for indoor high-precision 3D mapping and modeling [8,9]. By mounting the panoramic camera on the autonomous robot, the whole process can be completed without supervision or interference. In addition, the odometer on the robot is able to record the pose of each panoramic photo, which can also heighten the reliability and accuracy of 3D modeling [10]. The problem is how the mobile robot can automatically traverse the entire space in an indoor environment and find a collision-free path.

Pathfinding is one of the most complex and core domains in the research of mobile robots. With the increasing demand for automation and intelligence of mobile robots, in addition to remote control, robots are also required to automatically find collision-free paths, and even traverse completely in an unknown environment. The pathfinding or coverage in indoor environment consist even more problems since there are numerous narrow passages, doorways, and irregular static or dynamic obstacle [11].

Currently, the methods of the complete traversal pathfinding for unknown environments can be roughly classified into three categories.
(1)The first is the simple and random traversal. Under this strategy, the robot continuously runs forward, hits an object and then randomly turns an angle and continues to run forward to achieve compete traversal [12], but the method is too faulty to meet the needs in terms of benefit evaluation, such as working time, energy loss, and repeated coverage.(2)The second is to model the global environment after learning along the edge of environment, and to perform local pathfinding according to the existing environmental model, such as cell decomposition, grid method, and artificial potential field [13,14]. The cell decomposition method divides the global environment into several sub-regions, and each sub-region corresponds to a unique base point which represents this sub-region. The producer of complete traversal is transformed into finding a path on the interlaced network in the map model to traverse all the base points [14]. Similarly to cell decomposition, grid method divides the map into a series of grids. In this case, we can avoid complex calculations when dealing with obstacle boundaries. However, both of the two algorithms will be computationally intensive, when the area of the environment is too large [15]. Artificial potential field is a virtual force method proposed by Khatib [16]. Its basic idea is to regard the movement of the robot in the surrounding environment as an abstract motion in the artificial gravitational field. The target point produces “gravity” to the mobile robot, and the obstacle generates a “repulsive force” to the mobile robot to control the movement of the mobile robot. The path planned by the potential field method is generally smooth and safe. But the problem is the robot may get into an infinite loop or a local minimum.(3)The third is the walking pathfinding. The robot does not need to model the environment in advance. During the movement, the robot acquires real-time environmental information through on-board sensor, such as a camera or laser scanner [17], and uses an algorithm to find a local optimal direction according to the information. Then, the robot moves along the determined direction. The local path planning can be regarded as a procedure to find the next local optimal direction to proceed in. Because the local doesn’t need any priori information about the working environment, and the computation of this method will not increase as the area of the environment expands, it can avoid the problems of the two kinds of methods above and find high-resolution path in short time which is an important factor in practical application [18]. However, if only the local pathfinding is performed in a complex environment, there may be repeated traversals which will increase the time consumption and path length required for the complete coverage.

In the paper, a novel indoor mapping robot walking pathfinding detection algorithm is proposed. Like the walking pathfinding detection method, we utilize the laser scanner to acquire the information about real-time pose of the robot and scene map around at first. Then, according to the preset eight directions, we divide the scene map into eight areas and calculate the probabilistic path vector of each one. Next, we also propose a series of reasonable mechanisms, which can increase the efficiency and safety of the complete traversal greatly, to calculate the local optimal direction. Finally, aiming at the problem of the repeated traversal, we put forth a global pathfinding in addition to local pathfinding to eliminate the weaknesses of each method and strengthen the practicality in complex environment. Extensive experiments in three typical indoor environments, demonstrates that the proposed algorithm could be applied widely. And the discussion and conclusions are presented following.

The contributions of our work are summarized as follows:(1)An efficient solution for complete traversal in unknown indoor environment is proposed. The proposed algorithm is characterized by relatively less computation but high sophistication, which can ensure the robot to perform the real-time local pathfinding during the traversal.(2)Several mechanisms and feedback are proposed to determine the optimal local direction during local pathfinding, and increase the ability to avoid the obstacles automatically, including weight of rays, gray area exploration, automatic obstacle avoidance, motion direction inertia feedback and weight of feature pixels. The global pathfinding can reduce the time consumption and path length required for complete traversal without adding intensive computation to the algorithm, when the robot work in a complex environment.(3)Three typical indoor environments are tested precisely. The small-time consumption and short path length of the experiment verifies the efficiency of the proposed algorithm.

## 2. Eight-Direction Scanning Detection (eDSD) Algorithm

In the proposed eDSD algorithm, indoor mapping robot will be used to obtain scene map and real-time pose based on SLAM at first. Then, we divide the scene map into 8 areas and calculate the probabilistic path vector of each area. Considering the demand of efficient traverse, obstacle avoidance, and instantaneity in practical applications, we perform and optimize the local pathfinding, in terms of weight of rays, gray area exploration, automatic obstacle avoidance, motion direction inertia feedback and weight of feature pixels. Finally, we utilize the result of local pathfinding so as to realize the global pathfinding for indoor location environment of large-scale and complex structures. The flowchart of the proposed algorithm is illustrated in Figure 1. 

### 2.1. Simultaneous Localization and Mapping

Simultaneous localization and mapping is fundamental to the robotic pathfinding. During finding the path, the means for obtaining information and the method for processing the information vary greatly based on type of the sensor. Generally, they can be divided into laser-based SLAM [19,20] and vision-based SLAM [21,22]. Based on laser scanner, laser-based SLAM calculates distance information by actively emitting optical signals and calculating its propagation time. The advantage of this method include high accuracy of measurement, strong anti-interference ability and speed of acquiring data, although relatively little environmental information can be obtained and the errors will be accumulated by the odometer during long-term and large-scale moving. However, it is in line with the instantaneity and accuracy of the indoor pathfinding, and it can provide a two-dimensional plan of the environment. The latter is mainly based on RGB-D cameras, monocular, binocular or fisheye cameras. Generally, vision sensor can acquire larger amount of information, with lower costs, and more direct results. However, when the robot is in an unknown environment, object feature matching or simultaneous localization and mapping may not be completed well. Additionally, the vision-based method is computationally intensive, and it may be difficult to meet the demand for robot navigation with high instantaneity and high accuracy. Compared with the vision-based SLAM, the laser-based SLAM has already been very mature and popular for mobile robot navigation, especially where GPS doesn’t work. At present, lots of efficient methods, such as ICP, NDT, P_L_ICP [23,24,25], are used in laser scan matching, which plays a pivot role in laser-based SLAM [26]. And Andreas Nuchter even proposed cached k-d tree [27] to further accelerate the search for ICP algorithms. Therefore, we choose the laser-based SLAM to obtain scene map and real-time pose.

The SLAM algorithm can be summarized as follows: the robot moves from the starting position in an unknown environment, and it locates itself according to position estimation and map matching during the moving, and builds an incremental map based on its own localization [28]. Using the SLAM algorithm, you can directly subscribe to the real-time pose (position and direction) of the robot and the obstacle information around it under the corresponding topic. Figure 2 is a scene map of an office made with the Gmapping SLAM algorithm. This algorithm is currently the most widely used laser SLAM algorithm, which was first proposed by Murphy and Doucet et al. [29,30,31]. As it is shown in Figure 2, the scene map is divided into three parts: gray, white and black. The gray area represents the unknown area; The white area has no obstacle; The black area represents the obstacles.

### 2.2. Probabilistic Path Vector

In this paper, we propose to use an eight-direction scanning detection algorithm to process scene maps to obtain the probabilistic path vector of each of eight regions. The probabilistic path vector is composed of feature pixel, including the information of property and position of these feature pixel points. The property refers to the pixel value of the feature pixel, and the position refers to the coordinate of the feature pixel in the map coordinate system.

As shown in Figure 2, it is illustrated which areas are available, and which are obstructed. If the laser is blocked by obstacles during scanning, it will leave black pixels at the corresponding position of the map. Therefore, according to the principle of laser scanner, we develop a scheme for processing scene map.

We has set eights certain directions with which the robot can go along. They are up, down, left, right, top left, top right, bottom left, bottom right. There is 45° interval between every two adjacent directions. All of the eight directions are fixed relatively to the scene map, and they are not changed with the orientation of the robot. Figure 3 shows the model of the eight directions. Before each movement, the robot should select an optimal direction from these eight directions.

In addition to eight directions around the robot, there are eight areas around eight directions. Each area is composed of the ±15°
around the direction which is the center line of the area. Take the up side as an example for illustration.

Considering the effectiveness and efficiency of the algorithm, only 11 rays are sequentially emitted from the robot’s central point (j, i) in one area, and the interval between each two adjacent rays is 3° (Figure 4). It is the pixel lying on the ray that are defined as feature pixels. Feature pixel scanning is performed from the outside of the robot. When the ray passes through white pixel (with the value of 0) or gray pixel (with the value of −1), the next pixel along the ray continues to be analyzed. The ray stops developing forward when the boundary of the map or a black pixel (with the value of 100) is reached. 

Take the leftmost ray in the Figure 4 as an example to illustrate the process. “Δ” indicates the white pixel through which the ray passes. The ray eventually reaches a black pixel and stops developing forward. And the feature pixels on this ray is made up of these white pixels and the black pixel.

Finally, we can obtain a feature pixels set which consists of all of the feature pixels on the 11 rays and use the feature pixels set to represent the probabilistic path vector of the whole area. In this way, the environmental information around the robot can be effectively restored. 

We obtain the feature pixels set by Algorithm 1:


**Algorithm 1. Feature pixel point screening based on eight-point scan path detection**
Input: scene map and robot’s poseOutput: coordinates and pixel values of the set of feature pixels Calculate the robot’s coordinate (j, i) based on the scene map and robot’s pose;N←1;Δ There are 11 rays in total in one area;**while** N ≤ 11 **do** **for** y←i **to** 0 **by** −1 **do**  Calculate coordinate (x, y) according to the *N*th ray’s slope and robot’s coordinate (j, i);
  **if** ((x, y) in the map) **then**
    **if** (pixel value of (x, y) = 100) **then**      Record the coordinate and pixel value of feature pixel (x, y);      **exit**;    **else**
      Record the coordinate and pixel value of feature pixel (x, y);    **end if **  **else**    **exit**;  **end if**
 **end for**
 N←N+1;
**end while**


### 2.3. Local Pathfinding 

In this step, we calculate the reachable point (RP) of each area, according to the probabilistic path vectors. Then, we compare the values of RP of the eight directions and select the direction in which the RP is the largest. Then, the robot will move along with the chosen direction for a certain distance and perform the local pathfinding again. RP could be calculated by the following formula:(1)$$RP=\overset{10}{\underset{n=0}{\sum }}\sum \frac{weight_{xy}}{\sqrt{{\left(x-j\right)}^{\;}{}^{2}+{\left(y-i\right)}^{2}}}$$
where (j, i) is the coordinate of the center point of the robot in the map coordinate system. And (x, y) is the coordinate of the feature pixel selected on the nth ray.$\sqrt{{\left(x-j\right)}^{\;}{}^{2}+{\left(y-i\right)}^{2}}$ is equal to the distance between the robot and the feature pixel. $weight_{xy}$ is the weight of the feature pixel, including the weight of black feature pixel (wb), weight of white feature pixel (ww), and weight of gray feature pixel (wg). 

Then, in order to meet the demand of practical application, we should optimize the Equation (1) in several aspects. The following two points need to coincide in the process of the pathfinding. Firstly, pathfinding is supposed to eliminate the unknown area on the map as efficiently as possible. Secondly, it should ensure the robot to avoid collisions with obstacles. Therefore, we divide RP into three parts as Equation (2) to meet the requirement above, and calculate each part respectively.
(2)RP=Gray+Black+White
where Gray is calculated by only gray feature pixels in the whole feature pixels set, Black is calculated by only black feature pixels in the whole feature pixels set, White is calculated by only white feature pixels in the whole feature pixels set.

There are 5 mechanisms proposed in the following 5 chapters. Firstly, we assign each of the 11 rays in one area with different weights (in Section 2.3.1). Then, we adjust Gray,Black and White in Equation (2) to explore gray area (in Section 2.3.2), automatically avoid obstacle (in Section 2.3.3), and receive motion direction inertia feedback (in Section 2.3.4). Finally, we calculate wb, ww and wg based on the optimized formula (in Section 2.3.5).

#### 2.3.1. Weight of Rays

As mentioned above, the area around each direction is equivalent to the feature pixels set on 11 rays. However, the 11 rays themselves are not exactly equivalent. If the feature pixels on the rays of different inclinations have the same effect on the RP, there is a possibility of misjudgment. As shown in Figure 5, it is clear that the difference in the abscissa of the black pixels on the outer rays is greater than the width of the robot, and the robot can pass directly through this direction without being hit. However, because there are black pixels, which are relatively closer to the robot, on the outer six rays, the robot may not choose to go forward in this direction, according to the Equation (1). Similarly, there are lots of gray pixels detected on the outer 6 rays and relatively less black pixels on the inner 4 rays (Figure 6). It is possible that the robot will still select this direction, according only to Equation (1). These misjudgments will have negative influences on the efficiency of pathfinding. In order to eliminate the misjudgment, we firstly limit the angle of each area to 30° and leave some areas still in white (Figure 3). And the inner rays, which are closer to the eight directions, should be assigned greater weight than the outer rays. Thus, we assign different weights to the 11 rays, and the weights are consistent with Gaussian distribution. The equation of Gaussian distribution is presented as follow:(3)f(x)=e−x2100π
where *f*(*x*) represents the weight of the feature pixels on the xth ray. 

Therefore, the modified RP calculation formula is:(4)$$RP=\overset{10}{\underset{n=0}{\sum }}f\left(n-5\right)\sum \frac{weight_{xy}}{\;\sqrt{{\left(x-j\right)}^{\;}{}^{2}+{\left(y-i\right)}^{2}}}$$

Table 1 shows the weights of 11 rays in the positive direction as the 0th ray, 0th, ±1, ±2, ±3, ±4, ±5.

#### 2.3.2. Gray Area Exploration

The most basic requirement for mapping robot to completely traverse and create a map of the environment in an unknown environment is that the robot’s laser scanner can scan the entire environment. Thus, the robot must know which direction to go to explore more unknown areas. However, because the range of laser scanner is always within several meters, and the gray areas on the map will be correspondingly several meters away from the robot, if Gray (Equation (2)) is calculated according to Equation (4), the value of Gray will be so small that it can be almost ignored, compared with the value of Black and White. In this case, the robot will not be led to the unknown area efficiently. Therefore, we calculate Gray with Equation (5).
(5)$$Gray=\overset{10}{\underset{n=0}{\sum }}f\left(n-5\right)\sum \frac{wg}{\sqrt{{\left(x-x^{\prime }\right)}^{\;}{}^{2}+{\left(y-y^{\prime }\right)}^{2}}}$$
where wg is the weight of the gray feature pixel. (x, y) is equal to the coordinates of the gray feature pixel screened on the nth ray in the area. $\left(x^{\prime },y^{\prime }\right)$ represents a virtual point set along the direction of the area and closer to the gray feature pixels. Take the up, one of eight directions, as example. $\left(x^{\prime },y^{\prime }\right)$ can be calculated according to Equation (6).
(6)$$\{\begin{array}{c} x^{\prime }=j,\\ y^{\prime }=i-20\times \left|\frac{\sqrt{{\left(x-j\right)}^{\;}{}^{2}+{\left(y-i\right)}^{2}}}{20}\right| \end{array}$$

The distance between the feature pixels and robot can be substituted into the distance between the virtual point and the feature pixels. Even if the unknown area is far away from the robot, it will not be ignored.

#### 2.3.3. Automatic Obstacle Avoidance

In addition to gray area exploration, another requirement of the indoor complete traversal algorithm is to circumvent obstacles. In practical applications, the mapping robot may carry a panoramic camera to capture panoramic images in the environment, and combine the real-time pose of the robot with the panoramic images to reconstruct the three-dimensional environment. Therefore, the collision of the robot with the obstacles is likely to damage the instrument on the robot. In the eight-direction scanning detection algorithm, we calculate the value of RP in one area to represent the direction and compare RP among 8 directions. However, it cannot ensure that there must lie no obstacle in the direction we finally choose. For example, the situation, as shown in Figure 7, may occur. If the value of RP is calculated according to the Equation (4), it is possible to make this direction the best option, and if the robot goes forward along this direction, it will inevitably collide with obstacles.

In order to overcome the problem, we adjust the Equation (1) and calculate Black (Equation (2)) through Equation (7).
(7)$$Black=\{\begin{array}{c} \overset{10}{\underset{n=0}{\sum }}f\left(n-5\right)\sum \frac{wb}{\sqrt{{\left(x-j\right)}^{\;}{}^{2}+{\left(y-i\right)}^{2}}},\;(\sqrt{{\left(x-j\right)}^{\;}{}^{2}+{\left(y-i\right)}^{2}}>10)\\ \overset{10}{\underset{n=0}{\sum }}f\left(n-5\right)crash\times \sum \frac{wb}{\sqrt{{\left(x-j\right)}^{\;}{}^{2}+{\left(y-i\right)}^{2}}},\;\left(\sqrt{{\left(x-j\right)}^{\;}{}^{2}+{\left(y-i\right)}^{2}}\leq 10\right) \end{array}$$
where wb is equal to the weight of the black feature pixel. (x, y) is the coordinate of the black feature pixel on the nth ray, crash represents collision factor. By adding the collision factor to the calculation, when the distance between the robot and the black pixel is less than 10px, the value of Black increases sharply. crash can be calculated according to Equation (8).
(8)$$crash=\left(11-\sqrt{{\left(x-j\right)}^{\;}{}^{2}+{\left(y-i\right)}^{2}}\right)\times 20$$

In this way, it is ensured that the robot can be effectively braked before the collision. Also, the value of Black will not be so large that it dominates the value of RP, before the robot get close enough to the obstacles.

The processing of White is not changed, and the formula is as follows:(9)$$White=\overset{10}{\underset{n=0}{\sum }}f\left(n-5\right)\sum \frac{ww}{\;\sqrt{{\left(x-j\right)}^{\;}{}^{2}+{\left(y-i\right)}^{2}}}$$
where ww is equal to the weight of the white feature pixel. 

#### 2.3.4. Inertia Feedback of Motion Direction

In addition to the above two problems, the robot will encounter a lot of trouble in the field test. As shown in Figure 8, there are many seats in a typical indoor environment such as an office room. But it does not require the robot to enter each seat to traverse the entire room and create mapping, considering (1) the scanning range of the laser scanner larger than the depth of the seats, (2) little efficiency and necessity in terms of indoor 3D mapping and data acquisition. As the mapping robot I in Figure 8, it only need to run through the corridor to obtain enough data. However, a small unknown area may still exist at the corner in the seat because of the shape of the seat or the item placed inside. In this case, the robot’s pathfinding will be interfered inevitably. 

In addition, before the mapping robot II approaches the corner, it has inferred that there is no road ahead, and identified an unknown area in the top to right. If the robot turns to the upper right at the position, it is likely to get too close to the corner of the seat. Due to the complex structure indoor environment in the real world, we should keep the robot from being scratched by the seats corner on the side. 

In order to solve these problems, an inertia factor is introduced to the algorithm. Every time when the robot moves, the direction of the robot will be recorded and published to a topic as a feedback mechanism. The last direction is obtained before the next direction selection, and we add an extra value of inertia factor to the RP value in the same direction with the last one. Then, we compare it with the RP value in other directions. In this way, the robot can move as far as possible in the direction of the last movement. As shown as Figure 9, the mapping robot I and mapping robot II can avoid entering the seats or turning too early and drive directly to the front of the wall. And because of the collision factor, crash will offset the inertia factor value and robot will stop moving forward when the mapping robot is near the wall.

#### 2.3.5. Weight of Feature Pixels

In this paper, the determination of the three weight of feature pixel is based on the following principles: If the robot faces both the white area (area without obstacle) and the gray area (unknown area), it must give priority to the direction leading to the gray area. Therefore, we determine the relationship between wg and ww: |wg|>|ww|.There is no more than one black feature pixel on each ray. Thus, the number of black feature pixels in the feature pixels set is much smaller than gray feature pixels or white feature pixels. In order to ensure that the robot can avoid obstacles, we set the relationship among the three weights of feature pixels:  |wb|>|wg|>|ww|.

According to the basic relationship, multiple experiments were performed to find relatively suitable weights: wb=−10.0, wg=2.0, and ww=0.2. Based on these three weights, the value of RP of each direction can be finally determined, and the robot selects the direction in which the RP value is the largest as the movement direction.

### 2.4. Global Pathfinding 

Local pathfinding is a process of continuously calculating the local optimal direction and moving forward in the direction of the chosen direction. This method is suitable when the environmental structure is relatively simple and the environmental area is small. However, as the complexity and area of the environment increase, the robot will take many repetitive paths when it traverses the workspace, which will greatly reduce the efficiency of the algorithm. Therefore, we introduce a global pathfinding to solve this problem.

The process is illustrated as Figure 10.

When the robot is in position 1, it identifies that the Gray values are greater than the threshold value in up, left and right sides. Thus, the position 1 is pushed into the stack. Since the RP value in the up side is the largest one among the eight RP values, it proceeds upward.When the robot moves to position 2 through several local pathfinding, it identifies that the gray values of the up and the right are larger than the threshold value. Similarly, since the position 2 is pushed into the stack, it still moves upward.When the robot moves to position 3, there is no unknown area in the sight of the robot. Therefore, the coordinate of position 2 will be read and the robot will return to position 2 easily by using many existing algorithm which can find a collision-free path in short time.Because there is still unknown area on the right side of the robot, the robot will finally reach position 4.At position 4, because there is no unknown area in the sight of the robot, the coordinate of position 2 is read and the robot returns to position 2 again.At position 2, the robot does not detect any unknown area anymore. Therefore, position 2 is popped, the coordinate of position 1 is read, and the robot returns to position 1.

Since the coordinate of the point where the robot is located and the last pushed point, and the scene map between them are known, we can regard the process of returning to the previous pushed point as a path searching in a known situation. There are many ways to quickly and accurately search for the shortest path in a known scene, such as Dijkstra algorithm [32], best-fast-search (BFS) algorithm [33], A*(A star) algorithm and derivative algorithm of A* [34,35,36]. A* is the most popular choice in path search among them, because it can be used to search for the shortest path, can quickly guide itself with heuristics, and can be used in a variety of situations. Therefore, the A* algorithm is used in the global finding path to return the robot to the previous pushed point.

According to the process above, even if the environment is complicated, the global pathfinding can enable the robot to traverse completely in the unknown environment in quiet short path and avoid repeated traversal efficiently. This will greatly improve the practicability of the algorithm in complex environments.

## 3. Experiment and Discussion

In order to test the feasibility of the eight-direction scanning detection algorithm, we perform the experiment in three typical indoor environments, and the results are summarized and analyzed.

### 3.1. Experimental Platform and Sites

The experimental platform includes the Robot Operating System (ROS) system, the mobile robot (turtlebot3-waffle), and the 2D laser scanner (360 Laser Distance Sensor LDS-01), single board computer (Intel^®^ Joule™ 570×), control board (OpenCR1.0), etc. [37] (Figure 11).

The three experimental sites are the office room, small museum and apartment. As shown in Figure 12, Figure 13 and Figure 14, the U-shaped curve, the connected room, the T-junction, etc. are all typical spatial structures in the indoor environment with representative significance and experimental value.

### 3.2. Experiment and Assessment 

At the beginning of the experiment, the robot was set at one end of the room, the robot and timer were started simultaneously. During the movement, the robot ran in a straight line at a line speed of 0.1 m/s and turns at an angular velocity of 1.0 rad/s. The real-time pose was recorded at a frequency of 10 hz for the observation of the complete coverage path. We observed the process of mapping and pathfinding on the computer and stopped the robot and timer until the entire room was already traversed and a complete map of the room was constructed. Then, we compared the path planned by the eDSD with the preset ideal path which is drawn according to (1) the range of radar (160 mm–3500 mm) and the size of the room to ensure the radar can cover the entire room, (2) the requirement of indoor 3D mapping and data acquisition. And we calculated the length and the total time consumption of the planned path and preset ideal path respectively. The total time consumption of the ideal path can be worked out according to the path length and velocity. Finally, the path length and total time consumption in the two cases were compared, and the deviation value of the planned path was calculated according to Equations (10) and (11).
(10)$$D_{1}=\frac{s\prime -s}{s}\times 100\%$$
(11)$$D_{2}=\frac{t\prime -t}{t}\times 100\%$$
where $D_{1}\;$ indicates the deviation of planned path from the ideal path in terms of path length. The $D_{1}$ value is equal to the percentage of that the planned path’s length more than that of the ideal path, s′ indicates the length of planned path, and s indicates the length of ideal path. $D_{2}\;$ indicates the deviation of planned path from the ideal path in terms of total time consumption. The $D_{2}$ value is equal to the percentage of the planned path’s total time consumption exceeds that of the ideal path, t′ represents the total time consumption of planned path, and t represents the total time consumption of ideal path.

### 3.3. Results and Analysis

In Figure 15, Figure 16 and Figure 17, the blue dots show the planned path of the robot in the office room, small museum and apartment. The blue dot is the position of the robot every second. And ideal path of the robot is drawn in the red line. Table 2 and Table 3 demonstrate the comparison between planned path and ideal path. 

The biggest challenge in moving indoors is its complex environmental structure. Long and narrow corridors, no enough turning space, slender legs of tables and chairs, irregular obstacles, etc., will affect the judgment of the robot and increase the times of turning, and finally reduce the efficiency of the coverage. As the U-shaped curve shown in Figure 15, the space left for the robot to turn is even not much wider than the width of the robot, which has an impact on the turning; the numerous obstacles in the small museum (Figure 16) and apartment (Figure 17) also challenge the sensitivity of the laser scanner. However, the result shows that $D_{1}\;$ of three kinds of environment are 3.26%, 1.85% and 3.88% respectively and $D_{2}\;$ are 3.16%, 17.10% and 8.09% respectively. All of the deviations are kept in low level. What is more, the mobile robot found a collision-free path and achieved the complete coverage successfully in the three working environments. It is illustrated that the algorithm meets the requirements of efficiency and safety when the robot traverses completely in unknown indoor environment.

### 3.4. Discussion

According to the maps drawn, we can see the planned path and the ideal path are not much different in a single room, such as office room and apartment, with less turns. However, in the three-connected structure of a small museum (Figure 16), the inner structure is so complex that the robot had to make multiple turns during the pathfinding, which contributes directly to the obvious higher total time consumption of planned path. The possible reason is that the odometry data that this algorithm relies on, due to the impact of wheel slip and various errors, will cause that cumulative error of the odometry data obtained by estimating the velocity integral will become larger and larger, especially during the turns, deceleration and acceleration. 

Then, the accumulation of position errors (dead reckoning error) in the odometry during long-term and large-scale motion caused a deviation between the scene map and the actual environment [38,39]. For this problem, we would like to try other SLAM algorithm, such as Google’s Cartographer SLAM [40] instead of Gmapping SLAM. 

Further, the robot now can move in eight directions and the flexibility is relatively poor. If we increase the number of directions which the robot can move along with from 8 directions to 16 directions or even more directions, it may perform well in terms of obstacle avoidance. Thus, we plans to increase the number of preset directions in future research to find the most reasonable method.

## 4. Conclusions

In this paper, the problem of the indoor mapping robot pathfinding in unknown environments is discussed. Firstly, we introduce the process of simultaneous localization and mapping using laser SLAM algorithm. According to the real-time pose of the robot, we divide the scene map into 8 areas, and analyze the map by obtaining the probabilistic path vector of each area. Based on the demand of efficiency and safety of the application, we propose a local pathfinding, including assigning reasonable weights to each ray, setting virtual point to substitute robot coordinate to increase the proportion of Gray in RP value, proposing the collision factor to improve the safety of the experiment, adding the motion direction inertia factor to improve the efficiency of the local pathfinding, and the obtaining the appropriate weights for the feature pixels. Only few computations applied in each local pathfinding greatly increase the instantaneity and accuracy of the local pathfinding. We also propose global pathfinding for the unknown environment of large-scale and complex structures. It can effectively reduce the times of repeated traversal.

Field tests show that the path length of the planned paths do not exceed 4% of that of the ideal path, and the total time consumption of planned paths do not exceed 18% of that of the ideal path. The results ensure that the robot can efficiently traverse a relatively large space in an unknown environment, effectively avoid obstacles, and that it is, therefore, widely applicable to different environments. The contribution of this research has an important significance for improving the automation degree of the indoor space modeling work. It can be applied not only to construction of unknown environments, but also to the indoor 3D model construction by carrying a panoramic camera.

## Figures and Tables

**Figure 1 sensors-18-04254-f001:**
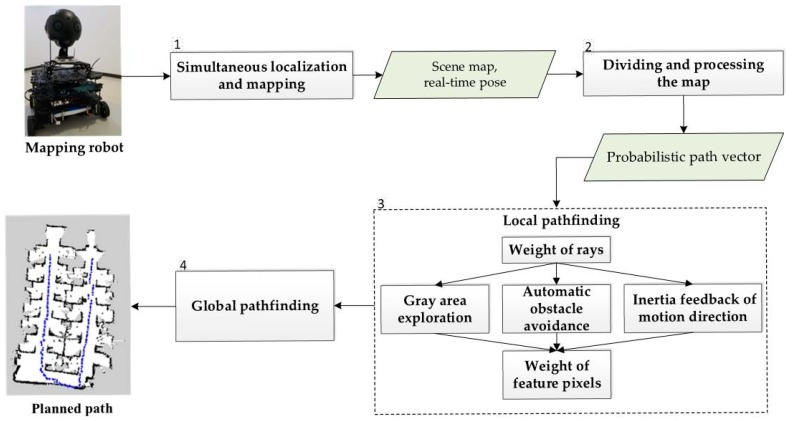
Overview of the eight-direction scanning detection algorithm.

**Figure 2 sensors-18-04254-f002:**
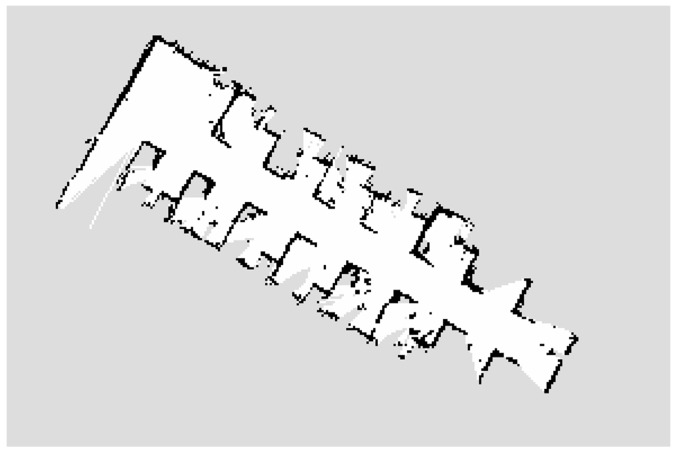
The scene map of the office made with Gmapping SLAM.

**Figure 3 sensors-18-04254-f003:**
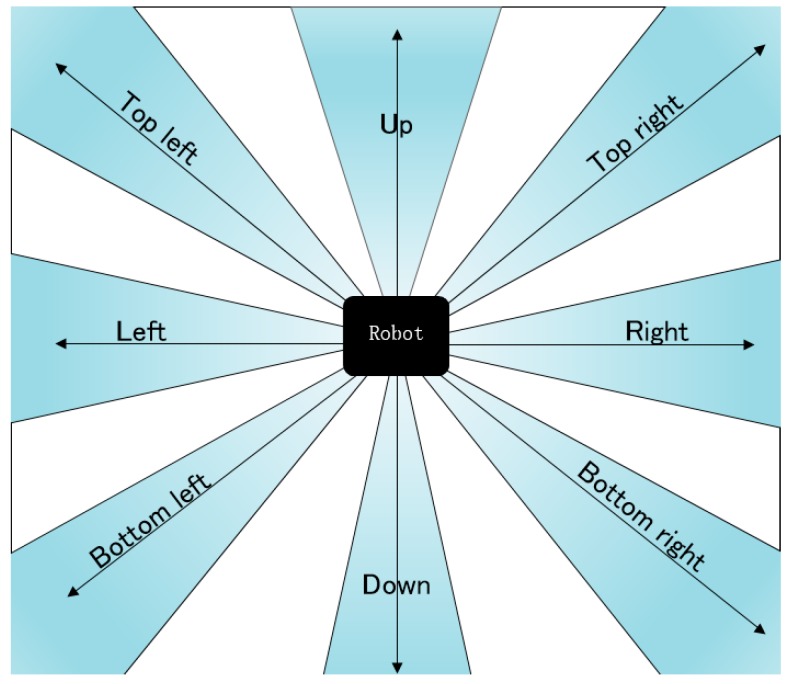
Schematic diagram of the eight-point scan path detection method.

**Figure 4 sensors-18-04254-f004:**
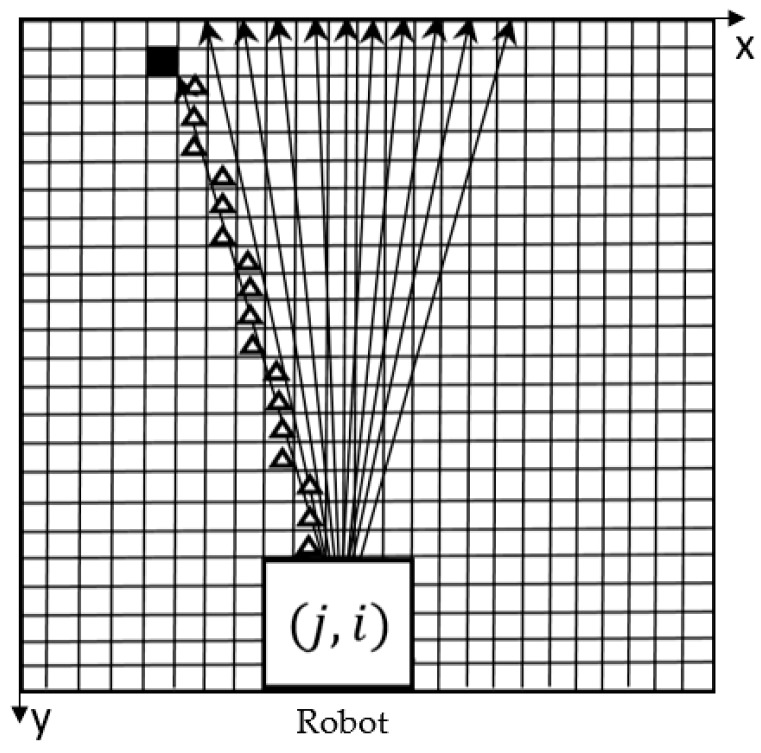
Schematic diagram of scanning principle (take the up side as an example).

**Figure 5 sensors-18-04254-f005:**
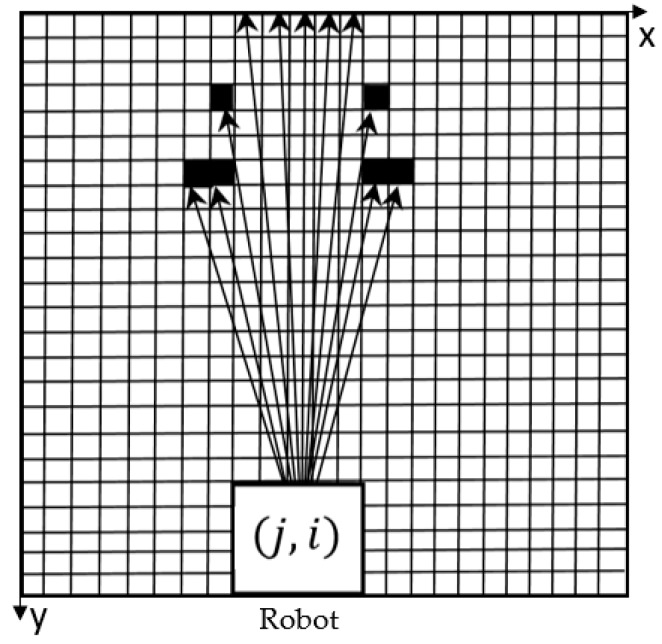
Schematic diagram of obstacles that may be encountered.

**Figure 6 sensors-18-04254-f006:**
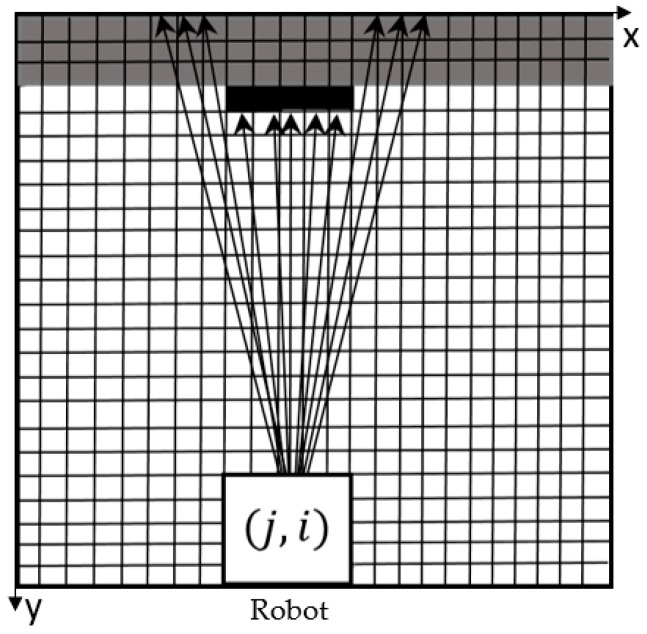
Schematic diagram of obstacles that may be encountered.

**Figure 7 sensors-18-04254-f007:**
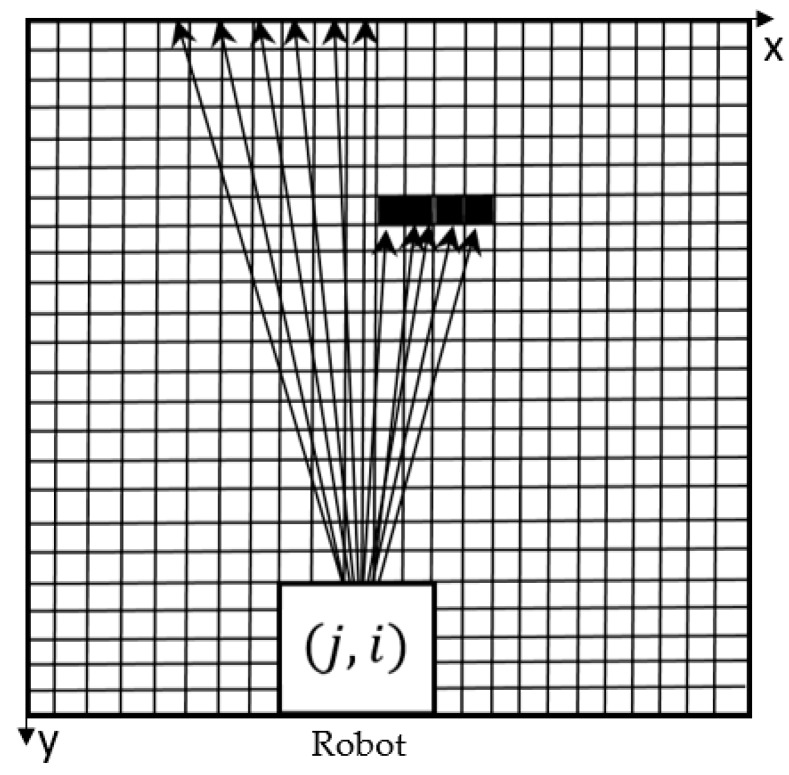
Schematic diagram of obstacles that may be encountered.

**Figure 8 sensors-18-04254-f008:**
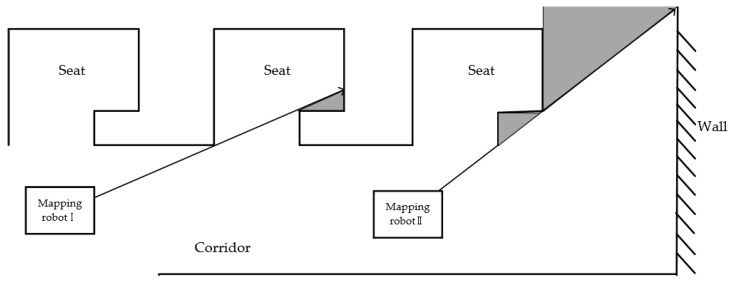
Schematic diagram of problem encountered during the actual scanning process.

**Figure 9 sensors-18-04254-f009:**
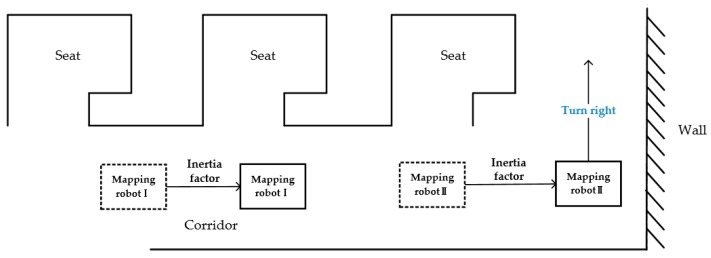
Theoretical path diagram after modifying the algorithm.

**Figure 10 sensors-18-04254-f010:**
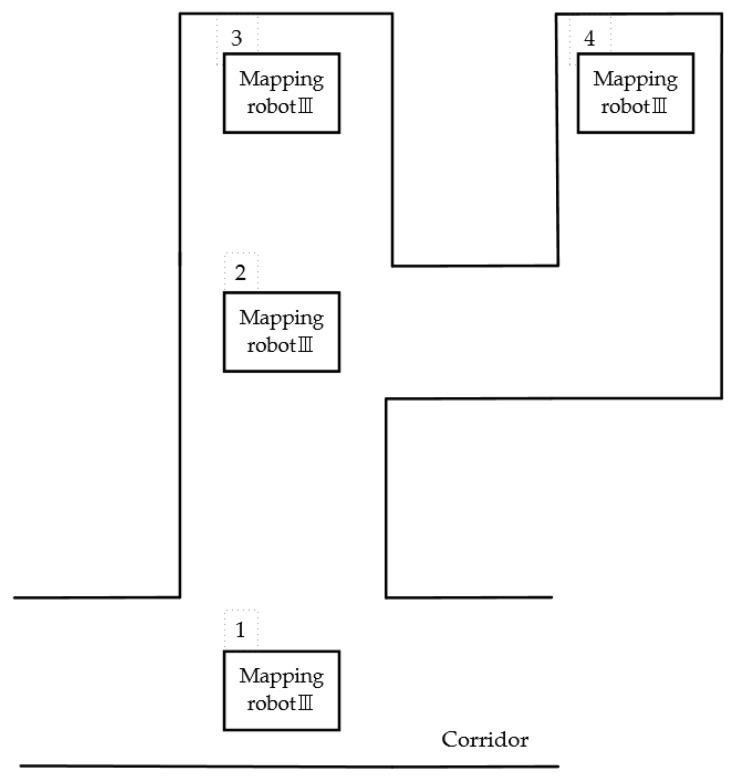
The process of global pathfinding.

**Figure 11 sensors-18-04254-f011:**
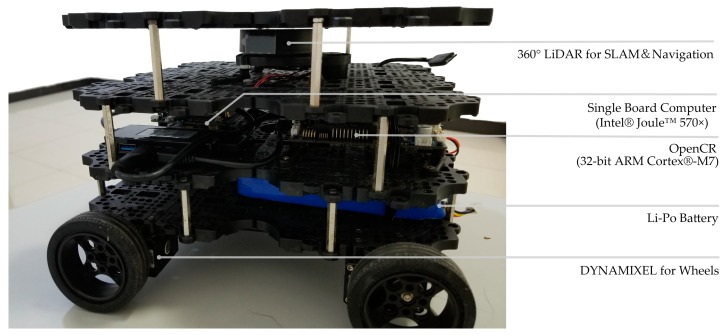
Experimental platform.

**Figure 12 sensors-18-04254-f012:**
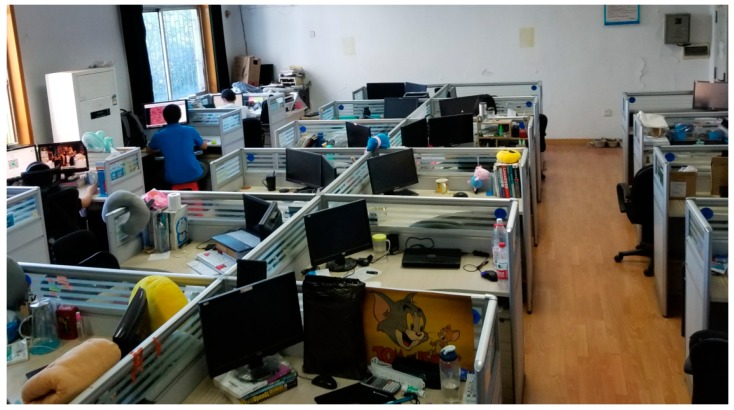
Office room.

**Figure 13 sensors-18-04254-f013:**
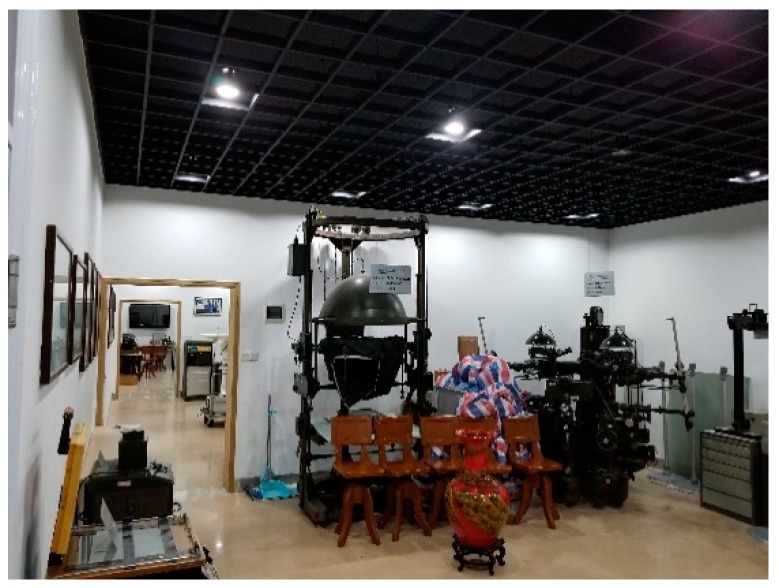
Small Museum.

**Figure 14 sensors-18-04254-f014:**
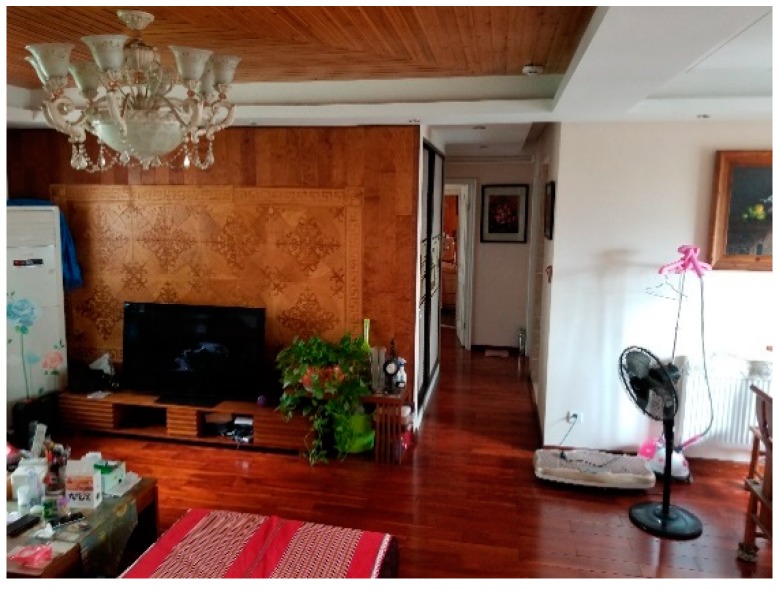
Apartment.

**Figure 15 sensors-18-04254-f015:**
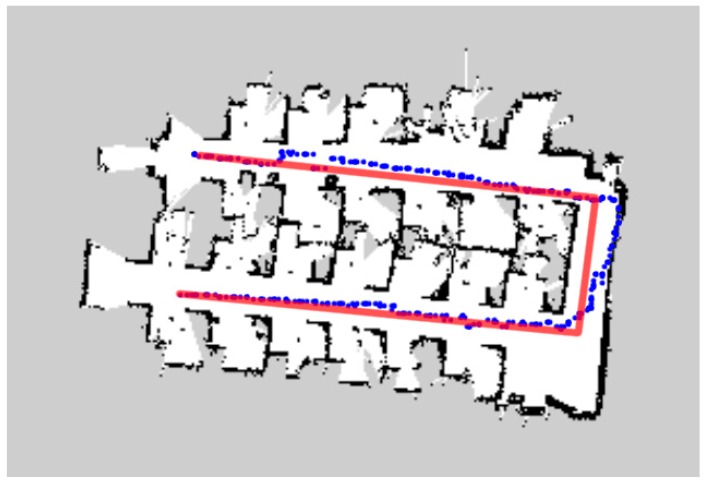
Planned and ideal path in the office room.

**Figure 16 sensors-18-04254-f016:**
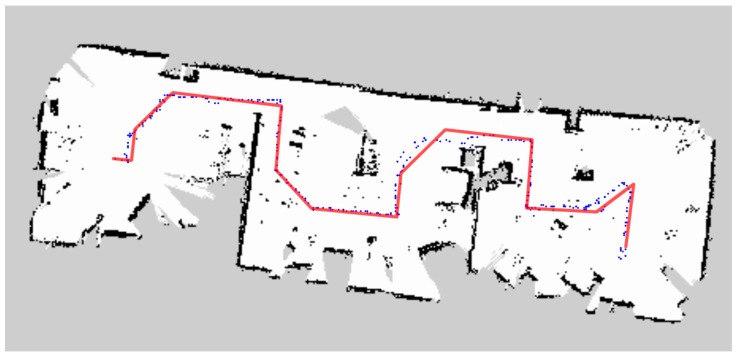
Planned and ideal path in small museum.

**Figure 17 sensors-18-04254-f017:**
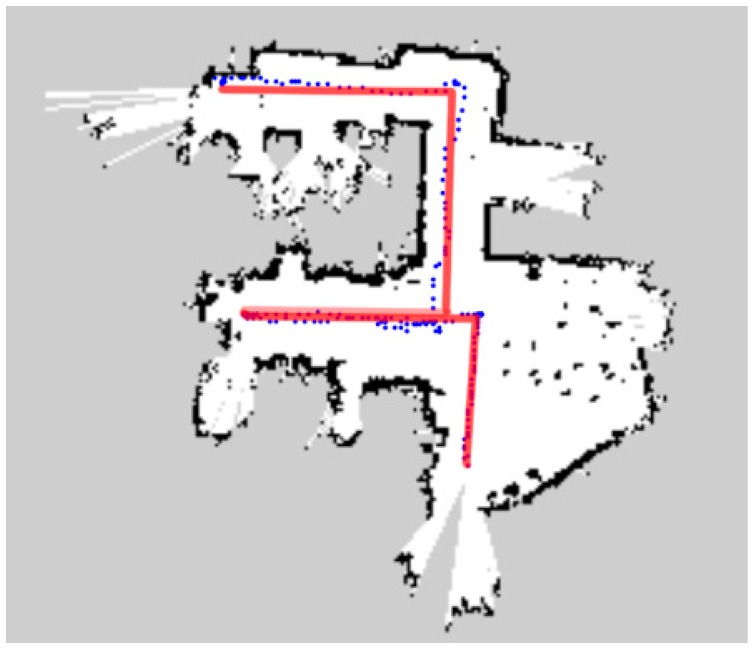
Planned and ideal path in apartment.

**Table 1 sensors-18-04254-t001:** Weights assignment of 11 rays.

No.	−5	−4	−3	−2	−1	0	1	2	3	4	5
**Weight**	0.456	0.605	0.754	0.882	0.969	1.00	0.969	0.882	0.754	0.605	0.456

**Table 2 sensors-18-04254-t002:** Comparison between path length of planned path and ideal path.

Indoor Environment	Ideal Path (m)	Planned Path (m)	$D_{1}\;(\%)$
Office room	21.76	22.47	3.26
Small museum	33.02	33.63	1.85
Apartment	18.31	19.02	3.88

**Table 3 sensors-18-04254-t003:** Comparison between total time consumption of planned path and ideal path.

Indoor Environment	Ideal Path (s)	Planned Path (s)	$D_{2\;}(\%)$
Office room	220.74	227.72	3.16
Small museum	345.9	405.06	17.10
Apartment	190.95	206.40	8.09

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
