# Peer review of "An Eight-Direction Scanning Detection Algorithm for the Mapping Robot Pathfinding in Unknown Indoor Environment"

_sensors, 2018, doi:10.3390/s18124254_

Round 1
Reviewer 1 Report
In the paper there is proposed an eight-direction scanning detection algorithm as a novel indoor mapping robot walking path finding algorithm to perform simultaneous localization and mapping in unknown environment. The laser scanner is utilized to acquire the information about real-time pose of the robot and scene map around at first. Then, according to the preset eight directions, the probabilistic path vector of the each area (8 certain areas around the eight directions) is calculated. The used approach is interesting and it is a good basis for further research.
Comments and questions:
1. Lines 15, 16 and 101: Missing space before parenthesis.
2. Line 220: Better specify 5 mechanisms. Are they these, which are described in chapters 2.3.1 – 2.3.5.?
3. Line 276: Maybe you mean Fig. 7.
4. Line 283, Eq.7: Is there condition in brackets? If yes, small font of brackets is for second condition.
5. Line 427 and 438: Spaces before and after line.
Author Response
Response to Reviewer 1 Comments
Point 1: Lines 15, 16 and 101: Missing space before parenthesis.
Response 1: Thank you for your suggestions. We have corrected the mistakes.
Point 2: Line 220: Better specify 5 mechanisms. Are they these, which are described in chapters 2.3.1 – 2.3.5.?
Response 2: Thank you for your comments. Yes, the 5 mechanisms are what we purposed in chapters 2.3.1-2.3.5. We have rewritten that part to make it more understandable.
Point 3: Line 276: Maybe you mean Fig. 7.
Response 3: Thank you for your comments. We did write Fig. 7 as Fig.6 wrongly. And it has been corrected.
Point 4: Line 283, Eq.7: Is there condition in brackets? If yes, small font of brackets is for second condition.
Response 4: Thank you for your comments. We have changed the brackets’ font.
Point 5: Line 427 and 438: Spaces before and after line.
Response 5: Thank you for your suggestions. We have corrected the mistakes.
Reviewer 2 Report
Nice piece of work and well presented, you definitely need to improve your reference list because in the present form it includes less than the 50% of the last 5 years. Try to include in your reference list articles like, ‘Design of an Autonomous Robotic Vehicle for Area Mapping and Remote Monitoring’, International Journal of Computer Applications, (ISSN: 0975 – 8887), Vol. 167, No 167, June 2017
Author Response
Response to Reviewer 2 Comments
Point 1: Nice piece of work and well presented, you definitely need to improve your reference list because in the present form it includes less than the 50% of the last 5 years. Try to include in your reference list articles like, ‘Design of an Autonomous Robotic Vehicle for Area Mapping and Remote Monitoring’, International Journal of Computer Applications, (ISSN: 0975 – 8887), Vol. 167, No 167, June 2017
Response 1: Thank you for your comments. We added 6 new references to our article.
Papoutsidakis, M., Kalovrektis, K., Drosos, C., & Stamoulis, G. (2017). Design of an Autonomous Robotic Vehicle for Area Mapping and Remote Monitoring. International Journal of Computer Applications, 167(12).
Lee, I. C., & Tsai, F. (2015). APPLICATIONS OF PANORAMIC IMAGES: FROM 720° PANORAMA TO INTERIOR 3D MODELS OF AUGMENTED REALITY. International Archives of the Photogrammetry, Remote Sensing & Spatial Information Sciences.
Nakagawa, M., Akano, K., Kobayashi, T., & Sekiguchi, Y. (2017). Relative Panoramic Camera Position Estimation for Image-Based Virtual Reality Networks in Indoor Environments. ISPRS Annals of the Photogrammetry, Remote Sensing and Spatial Information Sciences, 4, 349.
Hu, Q., Yu, D., Wang, S., Fu, C., Ai, M., & Wang, W. (2017). Hybrid three-dimensional representation based on panoramic images and three-dimensional models for a virtual museum: Data collection, model, and visualization. Information Visualization, 16(2), 126-138.Sun, T., Xu, Z., Yuan, J., Liu, C., & Ren, A. (2017). Virtual Experiencing and Pricing of Room Views Based on BIM and Oblique Photogrammetry. Procedia Engineering, 196, 1122-1129.
Moore, T., & Stouch, D. (2016). A generalized extended kalman filter implementation for the robot operating system. In Intelligent Autonomous Systems 13 (pp. 335-348). Springer, Cham.
Brossard, M., & Bonnabel, S. (2018). Learning Wheel Odometry and IMU Errors for Localization.
Reviewer 3 Report
The authors present an interesting path finding algorithm that can be implemented in autonomous robots. However, there are some questions that should be clarified in the article.
a) In the article the authors use 8 direction in which the scanning or the movement is performed. While at the first sight it may seem like a logical choice (most AI character in old computer games behave like this), in may not be an optimal solution in real word.
b) It would be useful, if the authors would describe the goal of the algorithm in details in the introduction (if I understand correctly, the goal is to map the area as efficiently as possible).
c) The authors state, that the proposed algorithm requires less computation, and is more efficient than traditional algorithms.
These days there are a lot of different platforms that can be used to control a robot, and these platforms have quite different processing power.
(c1) The first question is: what is the selected platform for controlling the robot?
(c2) How much processing power/time do the different portions of the algorithm require?
(c3) The available processing power is increasing at an exponential rate, for this reason I would say that we should optimize the algorithms for functionality and not processing speed.
d) If I understand well, the algorithm is using 11 virtual rays to scan the scene map provided by the SLAM algorithm. I think this is a brilliant idea. However, in this case there are not many limitations on what scanning can be performed. We could use more rays, and even omit the white areas showed in Fig. 3. The other possibility would be to consider the width of the robot. For example, in Fig. 5, the robot would know if it would fit through the black pixels or not.
e) It would be more useful, if the authors would describe their algorithm using pseudo-codes.
f) I am not sure if the Eq. (1) is correct. The (j,i) is the coordinate of the robot, not the feature pixel. Then why is the weight have ji indexes? The (x,y) are the coordinates of the feature pixel. This pixel may depend on the ray index, but this is not showed in the equation.
g) The authors use weighting of the rays to implement some kind of obstacle avoidance. I am sure that there can be a more efficient method (considering that we already have a SLAM map). In chapter 2.3.3 the authors introduce additional steps for obstacle avoidance, but I am still not sure if this would be an optimal solution. For example, on page 10, line 305 the authors describe a situation where the robot may collide with corners in the area. If true, it is a clear indication that the algorithm has some flaws.
To overcome the limitation, the authors introduce an “Inertia factor”. While this “Inertia factor” may be justified by the dynamic properties of the robot (for decreasing the need of deceleration and acceleration), using it as a path finding step is weird.
h) In Figs 15, 16, and 17 the authors visualize an “ideal” path. How was this path calculated? This path would still leave a lot of grey areas.
Author Response
Response to Reviewer 3 Comments
Point 1: In the article the authors use 8 direction in which the scanning or the movement is performed. While at the first sight it may seem like a logical choice (most AI character in old computer games behave like this), in may not be an optimal solution in real word.
Response 1: Thank you for your comments. We have considered the question of how many directions will be reasonable. Increasing the robot's movable direction does make the robot move more flexible, but it also has potential risks. In the real world, the scene plan of the robot we obtained is made by the GMapping SLAM algorithm, which is one of the most widely used algorithms in robot real-time localization and mapping. The odometer data that this algorithm relies on, due to the impact of wheel slip and various errors, will cause that cumulative error of the odometer data obtained by estimating the velocity integral will become larger and larger. If more directions for the robot to advance are provided simultaneously, it means that the times of the turns will increase, resulting in a larger accumulated error and finally great negative influence on path finding. And in fact our algorithm can satisfy the test in the selected environments, and more directions may get relatively poor results. In addition, this method is only a prototype, and we do have plans to increase the number of preset directions in future research to find the most reasonable method.
We have added the above explanation in the discussion part.
Point 2: It would be useful, if the authors would describe the goal of the algorithm in details in the introduction (if I understand correctly, the goal is to map the area as efficiently as possible).
Response 2: Thank you for your comments. We feel sorry that we didn’t state the goal clearly in the article. SLAM map is the by-product of this research. The goal of the research is to solve the inconvenience in the 3D reconstruction, data acquisition and measurement of closed indoor environment. The autonomous robot can serve as an intelligent panoramic acquisition platform for indoor high-precision 3D modeling, instead of two-dimensional mapping.
Two characteristics of the mobile robot can just heighten the efficiency and precision of measurement and data acquisition. 1) Mapping robot can work without supervision of any researcher in an indoor environment. 2) The odometer is able to provide the pose of the equipment at each moment.
We have rewritten the introduction and clarify our goal.
Point 3: The authors state, that the proposed algorithm requires less computation, and is more efficient than traditional algorithms.
These days there are a lot of different platforms that can be used to control a robot, and these platforms have quite different processing power.
1) The first question is: what is the selected platform for controlling the robot?
2) How much processing power/time do the different portions of the algorithm require?
3) The available processing power is increasing at an exponential rate, for this reason I would say that we should optimize the algorithms for functionality and not processing speed.
Response 3: Thank you for your suggestions.
1) TurtleBot3 Waffle is equipped with a single board computer (Intel® Joule™ 570x) and control board (OpenCR1.0) which is powerful enough to control not only DYNAMIXELs but also ROBOTIS sensors that are frequently being used for basic recognition tasks in cost effective way. The control board is open-sourced in hardware wise and in software wise for ROS communication.
We have corrected some mistakes in the section 3.1, and replaced a picture to show more details about the robot.
2) It is too difficult to calculate the processing power/time which different portions of the algorithm require, but the processing time to find the path each time is available. The algorithm takes about 50ms each time from calculating probabilistic path vector (most of time is consumed in this step) to the final determination of the forward direction, and the frequency of subscribing the pose of the robot from odometer is 10 Hz (i.e. the environment information is read in 100ms once). Therefore, this processing speed can basically guarantee that the path can be obtained once every time data is obtained.
3) We are in favor of your point of view. We tried our best to improve the functionality of the algorithms, and then made the working efficiency as high as possible simultaneously. And we will improve its function in more aspects in the future research.
Point 4: If I understand well, the algorithm is using 11 virtual rays to scan the scene map provided by the SLAM algorithm. I think this is a brilliant idea. However, in this case there are not many limitations on what scanning can be performed. We could use more rays, and even omit the white areas showed in Fig. 3. The other possibility would be to consider the width of the robot. For example, in Fig. 5, the robot would know if it would fit through the black pixels or not.
Response 4: Thank you for your suggestions.
The reason for leaving the white part in Figure 3 is that we intentionally reduce the influence of the areas away from the eight directions on the robot path finding. Attributing different weights to different rays is for the same reason as well. In chapter 2.3.1, the rays closer to the center ray (forward direction) should have a larger weight. We have added the explanation in 2.3.1.
The width of the robot is a very important factor when the experiment is done in the real world. We had tried to compare the width of the robot and the interspace between the black pixels to ensure whether it would fit through. However, we found it difficult to calculate the length of the interspace. In fact, what we can obtain are a set of the feature pixels and their poses. Then, we should do classification processing, many distance calculation and projection operations to find out whether the shortest length of the interspace is wilder or narrower than that of robot. And the algorithm purposed in chapter 2.3.1 can meet the requirement. Thus, we abandoned this plan.
Point 5: It would be more useful, if the authors would describe their algorithm using pseudo-codes.
Response 5: Thank you for your suggestions. This algorithm in section 2.2 has been rewritten in pseudo-codes.
Point 6: I am not sure if the Eq. (1) is correct. The (j,i) is the coordinate of the robot, not the feature pixel. Then why is the weight have ji indexes? The (x,y) are the coordinates of the feature pixel. This pixel may depend on the ray index, but this is not showed in the equation.
Response 6: Thank you for your comments. Actually, we make a serious mistake here. We had intended to use weightxy to represent the weight of the pixel at (x, y). And the value of the weightxy must be equal to one of the three weights- the weight of gray, the weight of white and the weight of black. For example, if the pixel at (x, y) is a white feature pixel, then weightxy = ww (weight of white feature pixel).
We have corrected Eq. (1) and Eq. (4).
Point 7: The authors use weighting of the rays to implement some kind of obstacle avoidance. I am sure that there can be a more efficient method (considering that we already have a SLAM map). In chapter 2.3.3 the authors introduce additional steps for obstacle avoidance, but I am still not sure if this would be an optimal solution. For example, on page 10, line 305 the authors describe a situation where the robot may collide with corners in the area. If true, it is a clear indication that the algorithm has some flaws.
To overcome the limitation, the authors introduce an “Inertia factor”. While this “Inertia factor” may be justified by the dynamic properties of the robot (for decreasing the need of deceleration and acceleration), using it as a path finding step is weird.
Response 7: Thank you for your suggestions.
First of all, the SLAM map is being drawn while the robot is finding the path, it was not obtained from the beginning. If we have a SLAM map from the beginning, there are many ways to plan a more perfect path.
And I have to concede that this problem is too complex, because the real world situation is not as simple as the simulation. Although we cannot say the strategy in chapter 2.3.3 is the most optimal, it can actually keep the robot from crashing into the obstacles. However, 2.3.3 is passive for the collision on the side. As is known to all, cars are more likely to be scratched than rear-end collided because it's hard to predict whether the side of the car will touch obstacles as people drive forward, especially when the car is about to get out of a narrow alley and turn right or left. The inertia factor just has the ability to prevent the robot from being scratched by the seat corner. Apart from this reason, there are other reasons to introduce the inertia factor. For example, considering the efficiency and necessity in terms of indoor 3D mapping and data acquisition, it is unnecessary to go in to the seats and map the radar blind angle as described in Fig. 8. Drawing a perfect map is not the final goal. Thus, we propose that this inertia factor plays a decisive role in path finding.
We have made some modification and add some explanation in the 2.3.4.
Point 8: In Figs 15, 16, and 17 the authors visualize an “ideal” path. How was this path calculated? This path would still leave a lot of grey areas.
Response 8: Thank you for your suggestions. We still take the application of indoor 3D modeling and data acquisition into consideration when making the ideal path. This ideal path is an optimal path for the robot to take a panoramic camera to obtain an indoor panoramic image efficiently and safely. There is no need to get a perfect slam map, which itself is just a by-product of this path finding process. And we intended to use the map to make the route, and the experimental results more expressive.
Then, we also considered the range of the robot radar (from 120mm to 3500mm) and the size of the room. Moving along the route, the radar can cover the entire space theoretically.
We have added the explanation about the ideal path in section 3.2.
Round 2
Reviewer 3 Report
The authors answered all my questions.